# Adaptively Private Next-Token Prediction of Large Language Models

## Abstract

As Large Language Models (LLMs) proliferate, developing privacy safeguards for these models is crucial. One popular safeguard involves training LLMs in a differentially private manner. However, such solutions are shown to be computationally expensive and detrimental to the utility of these models. Since LLMs are deployed on the cloud and thus only accessible via an API, a Machine Learning as a Service (MLaaS) provider can protect its downstream data by privatizing the predictions during the decoding process. However, the practicality of such solutions still largely lags behind DP training methods. One recent promising approach, Private Mixing of Ensemble Distributions (PMixED) Flemings et al. (2024), avoids additive noise by sampling from the output distributions of private LLMs mixed with the output distribution of a public model. Yet, PMixED must satisfy a fixed privacy level for a given number of queries, which is difficult for an analyst to estimate before inference and, hence, does not scale. To this end, we relax the requirements to a more practical setting by introducing Adaptive PMixED (`AdaPMixED`), a private decoding framework based on PMixED that is adaptive to the private and public output distributions evaluated on a given input query. In this setting, we introduce a noisy screening mechanism that filters out queries with potentially expensive privacy loss, and a data-dependent analysis that exploits the divergence of the private and public output distributions in its privacy loss calculation. Our experimental evaluations demonstrate that our mechanism and analysis *can reduce the privacy loss by* $16\times$ while preserving the utility over the original PMixED. Furthermore, performing 100K predictions with `AdaPMixED` still achieves strong utility and a reasonable data-dependent privacy loss of $\epsilon = 5.25$.

## 1 Introduction

Large Language Models (LLMs) have accomplished huge commercial success due to their ability to memorize factual knowledge within their model parameters Petroni et al. (2019); Roberts et al. (2020). However, this memorization poses a serious privacy leakage vulnerability as LLMs are susceptible to training data extraction attacks on memorized text Carlini et al. (2021; 2019; 2022). The widely adopted approach to reduce memorization is to guarantee Differential Privacy (DP) Dwork (2006), a mathematical privacy notion, during the training of an LLM by applying strategic noise to per-sample gradients, called DP-SGD Abadi et al. (2016). Recent works have demonstrated that public pre-training followed by private fine-tuning substantially boosts the performance of DP LLMs Li et al. (2021); Yu et al. (2021a); Li et al. (2022); Ganesh et al. (2023). But the potentially large additive noise by DP-SGD, which scales proportionally to the total number of parameters of an LLM Kamath (2020), and the underutilization of ML accelerated hardware due to per-sample gradient calculations Yousefpour et al. (2021), have delayed wide-scale adoption of DP-SGD to LLMs.

Perhaps the biggest discrepancy between DP-SGD and commercial deployments of LLMs is that many Machine Learning as a Service (MLaaS) providers only provide API access to their LLMs, e.g. OpenAI's Chat-GPT. However, DP-SGD assumes an adversary has access to the gradients of an LLM at every iteration of training. This discrepancy leads to an overestimated privacy loss bounds Nasr et al. (2021) when considering an adversary with only black-box access to the model. Alternatively, DP prediction methods Dwork & Feldman (2018) offer potential improvement in model utility by explicitly exploiting this relaxation; however, in practice, DP prediction does not necessarily lead to higher utility than DP-SGD van der Maaten & Hannun (2020).

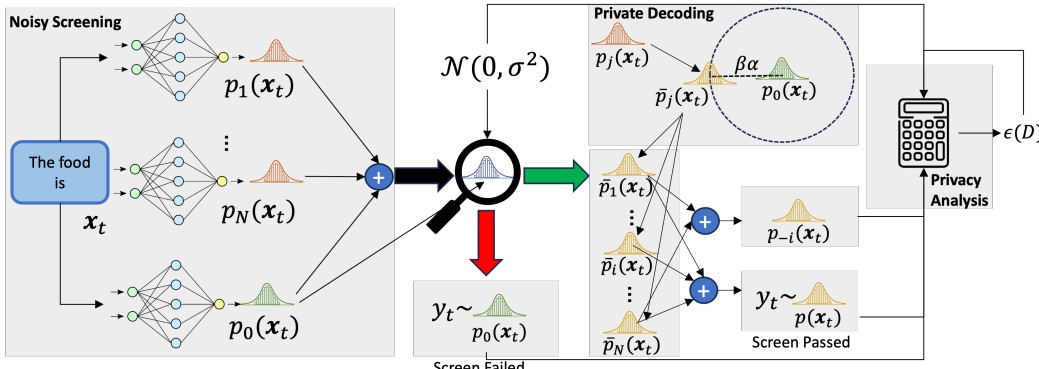

Figure 1: A brief overview of `AdaPMixED`. For each query **x** received by a user, noisy screening is performed by first mixing each private distribution $p_i(\mathbf{x})$ with the public distribution $p_0(\mathbf{x})$. Then the mixed distribution is privatized with Gaussian noise and is compared with the public distribution $p_0(\mathbf{x})$. If the screening fails, then the next token is sampled from the public distribution $y \sim p_0(\mathbf{x})$. Otherwise, private decoding is performed using PMixED. Our privacy analysis tracks the privacy loss $\epsilon(D)$ by analyzing the change of the output distribution when removing one model $p_{-i}(\mathbf{x})$ from decoding with PMixED for every query.

A recent promising DP prediction algorithm, called Private Mixing of Ensemble Distributions (PMixED) Flemings et al. (2024) is able to achieve higher utility than DP-SGD for certain query budgets by exploiting two crucial features: (1) There is an *inherent privacy* when generating the next token by sampling from the output probability distribution of a language model; (2) The privacy loss of a prediction can be bounded by employing a public model to mix the output distributions of privacy-sensitive LLMs. These observations allow PMixED to avoid additive noise and thus preserve model utility Flemings et al. (2024). However, the DP Prediction definition Dwork & Feldman (2018) that PMixED operates under is too rigid, as one must estimate a pre-specified privacy budget that will be satisfied for a certain number of predictions before querying an LLM, which is difficult. Moreover, this rigid requirement is satisfied by analyzing the privacy loss for each prediction without consideration for the output distributions of the private and public models, often leading to overly pessimistic privacy loss when the actual privacy loss is substantially smaller. Thus, PMixED, in practice, exhausts its privacy budget quickly and does not scale well to current generative applications.

In this work, we address the aforementioned limitations by introducing Adaptive PMixED (`AdaPMixED`), which makes the following contributions: (1) We provide a tighter privacy analysis of PMixED by leveraging the output distributions from auto-regressive LLMs to directly calculate the privacy loss. (2) We adopt a noisy screening mechanism only found in discriminative tasks Papernot et al. (2018); Zhu et al. (2020) to the language modeling setting, by using the Renyi Divergence to measure the distributional difference between projected and public distributions then filter out queries with large divergence. (3) We present a privacy and utility analysis of `AdaPMixED` and experimentally demonstrate improved privacy-utility tradeoff over PMixED. In particular, on the WikiText-103 dataset for a given number of queries, `AdaPMixED` can improve the utility by 1.4 perplexity over PMixED with 16 times less privacy loss. Furthermore, we demonstrate our method can practically scale to large number of queries, achieving a privacy loss of $\epsilon = 5.248$ for almost 100,000 predictions, while improving the utility over DP-SGD at a privacy level of $\epsilon = 8$ by nearly 2.5 perplexity. This is the first work to demonstrate that a DP prediction-based scheme can be practical for such extremely large queries, as prior work has evaluated on only at most 1k queries Flemings et al. (2024); Ginart et al. (2022); Zhu et al. (2023). We hope these promising results will spark further improvement in private decoding for real-world deployment of large-scale applications.

## 2 RELATED WORKS

Most of the progress in the differential privacy and LLM space has focused primarily on differentially private fine-tuning McMahan et al. (2017); Yu et al. (2021a); Li et al. (2021); Flemings & Annavaram

(2024). In many of these works, a pre-trained model trained on public data is further fine-tuned on a private downstream dataset $D$ using DP-SGD. The resulting model parameters are differentially private with respect to $D$. Private prediction methods Dwork & Feldman (2018) offer an alternative way to guarantee DP for LLMs, during predictions rather than training. Although some progress has been made in DP prediction methods for discriminative tasks Zhu et al. (2020; 2023), most DP prediction methods for generative LLMs produce substantial utility degradation Majmudar et al. (2022). Another body of work also explored DP prediction of LLMs without fine-tuning of private data Tang et al. (2023); Duan et al. (2024); Xie et al. (2024); Wu et al. (2023). In these works, generated prompts containing private information are used by an LLM via an API call. Our work differs from these in that we utilize LLMs fine-tuned on private data to generate DP predictions.

Our work relies on the PMixED framework Flemings et al. (2024), however, we pointed out compelling shortcomings of PMixED that limit its practicality in real-world applications. Another conceptually similar, but orthogonal approach, to PMixED is PATE Papernot et al. (2016; 2018), which also uses an ensemble of models trained on pairwise disjoint subsets of a private dataset to generate DP labels. However, PATE uses the Gaussian/Laplacian mechanism to perturb vote count histograms while PMixED relies on sampling and mixing private and public output distributions. Consequently, adopting PATE to the language modeling setting produces suboptimal results, since perturbing the aggregated output distribution of the teachers incurs large noise due to its large dimensionality Ginart et al. (2022). The output space can be truncated, as done in Tian et al. (2022), but this can cause an error in the perplexity calculation. Instead, truncated decoding schemes must use generation perplexity Fan et al. (2018); Holtzman et al. (2019); Pillutla et al. (2021), which does not apply in our experimental setup.

Although the high level goal of our noisy screening mechanism and data-dependent analysis is motivated by PATE Papernot et al. (2018), our approach to both differs significantly from PATE. PATE's noisy screening checks if the noisy max meets a threshold, whereas we use Renyi Divergence to measure the distributional difference between projected and public distributions to guide the screening process. This use of Renyi Divergence for screening is a novel aspect of our work, as no prior work has done this. Moreover, our data-dependent analysis iteratively examines neighboring ensembles to find the largest privacy loss, utilizing the probability distributions generated. On the other hand, PATE, designed for discriminative tasks, uses histograms to calculate class count differences as its data-dependent analysis. Our data-dependent analysis closely follows from Ginart et al. (2022).

## 3 Preliminaries

We start by reviewing some preliminaries on differential privacy, and then we will briefly discuss an overview of the PMixED framework. We will denote by $D = (d_1, ..., d_n)$ to be the private dataset and $D_{-i} = (d_1, ..., d_{i-1}, d_{i+1}, ..., d_n)$ to be the private dataset after removing point $d_i$. Our work focuses on next-token prediction, where given an input query $\mathbf{x} = x_1, x_2, ..., x_t$, which is a string of tokens from some vocabulary $V$, we choose the next token involves sampling from this probability mass function to obtain a token $\hat{x}_{t+1} \sim p(x_{t+1}|\mathbf{x}_t)$.

### 3.1 Differential Privacy and Renyi Differential Privacy

**Definition 3.1** (Approximate DP Dwork et al. (2014); Feldman & Zrnic (2021)). Let $\epsilon > 0, \delta \in [0, 1]$. A randomized algorithm $A : \mathcal{D} \to \mathcal{R}$ satisfies $(\epsilon, \delta)$-DP if for all datasets $D \in \mathcal{D}$ and any set of subset of outputs $S \subseteq \mathcal{R}$ it holds that:

$$\Pr[A(D) \in S] \leq e^\epsilon \Pr[A(D_{-i}) \in S] + \delta \text{ and } \Pr[A(D_{-i}) \in S] \leq e^\epsilon \Pr[A(D) \in S] + \delta$$

Another privacy notion of DP called Renyi Differential Privacy (RDP) contains convenient composition properties $(\epsilon, \delta)$-DP Mironov (2017).

**Definition 3.2** (Renyi Divergence Mironov (2017)). For two probability distributions $P$ and $Q$ defined over $\mathcal{R}$, the Renyi divergence of order $\alpha > 1$ is

$$D_\alpha(P||Q) = \frac{1}{\alpha-1} \log \mathbb{E}_{x \sim Q} \left[ \left( \frac{P(x)}{Q(x)} \right)^\alpha \right], \tag{1}$$

and define $D_\alpha^\leftrightarrow(P||Q) = \max\{D_\alpha(P||Q), D_\alpha(Q||P)\}$.

**Definition 3.3** (($\alpha, \epsilon$)-RDP Mironov (2017); Feldman & Zrnic (2021))**.** A randomized algorithm $A : \mathcal{D} \to \mathcal{R}$ is ($\alpha, \epsilon$)-RDP if for all datasets $D \in \mathcal{D}$ it holds that $D_\alpha^\leftrightarrow(A(D)||A(D_{-i})) \leq \epsilon$.

In certain cases, it is possible to further reduce the privacy loss $\epsilon$ by exploiting properties of the private dataset $D$ during the privacy analysis. We define data-dependent RDP below.

**Definition 3.4** (Data-Dependent RDP Papernot et al. (2018))**.** A randomized algorithm $A : \mathcal{D} \to \mathcal{R}$ is ($\alpha, \epsilon(D)$)-RDP if for all datasets $D \in \mathcal{D}$ it holds that $D_\alpha^\leftrightarrow(A(D)||A(D_{-i})) \leq \epsilon(D)$.

We discuss the implications of Data-Dependent RDP. Firstly, the privacy loss $\epsilon(D)$ can only be known once the computation of $A$ has halted. The release of $\epsilon(D)$ results in privacy leakage and thus must be privatized, which can be done by employing the smooth-sensitivity framework Nissim et al. (2007) as done by Papernot et al. (2018). Privacy amplification by subsampling (Theorem A.4), a property that strengthens the privacy guarantee of a DP algorithm, cannot be used concurrently with data-dependent accounting, as it is currently an open problem Zhu et al. (2020). Lastly, checking that a data-dependent privacy loss $\epsilon(D)$ does not exceed a pre-specified privacy budget $\epsilon_G$ leaks additional privacy Redberg et al. (2023), and hence, is incompatible with the DP prediction definition Dwork & Feldman (2018).

### 3.2 The PMixED Framework

PMixED Flemings et al. (2024) is a private decoding framework that guarantees DP at prediction time for generative LLMs. PMixED partitions a private dataset into $N$ pairwise disjoint subsets $D_i$, each of which is fine-tuned by an LLM $p_i$. Then during inference, each fine-tuned model generates its output probability distribution $p_i(\mathbf{x}_t)$, containing private information, on some input $\mathbf{x}_t$ received by a user. Additionally, the output distribution from a public model $p_0(\mathbf{x}_t)$ is obtained. PMixED then projects each of the private distributions onto a Renyi Divergence ball centered at the public distribution with radius $\beta\alpha$. The optimal projection is selected by choosing $\lambda_i$ such that $\lambda_i \leftarrow \arg\max_{\lambda \in [0,1]}\{D_\alpha^\leftrightarrow(\lambda p_i(\mathbf{x}_t) + (1-\lambda)p_0(\mathbf{x}_t)||p_0(\mathbf{x}_t)) \leq \beta\alpha\}$. Lastly, the projected distributions are averaged and then sampled to generate the next token. A precise description of PMixED can be found in Algorithm 2. We state the main privacy analysis results of Flemings et al. (2024) here and leave the details in Appendix C.

**Theorem 3.1.** Given $\beta, N$ chosen by the analyst, PMixED satisfies ($\alpha, \epsilon(\alpha, \beta, N)$)-RDP for some query $\mathbf{x}$ where

$$\epsilon_{\text{PMixED}}(\alpha, \beta, N) \leq \frac{\log\left(\frac{N-1+\exp(4\beta\alpha(\alpha-1))}{N}\right)}{\alpha-1} \tag{2}$$

**Theorem 3.2.** PMixED $\mathcal{P}$ is an ($\alpha, \epsilon_G$)-RDP prediction algorithm with respect to $D$ for a query budget $T$.

Although PMixED has been shown to satisfy the private prediction definition (Theorem 3.2), this limits PMixED to a fixed privacy level for only a certain number of queries. Once the query budget is exhausted, either no more predictions can be performed, or the privacy guarantee will decay with each additional prediction. For real-world LLM inference applications where it's impossible to know the total number of queries ahead of time, this places a huge burden on the analyst to hypothesize an adequate query budget and privacy level before deployment. Moreover, the privacy loss between different queries can vary widely, yet it is not considered when pre-specifying a fixed privacy level.

## 4 AdaPMixED: An Adaptive Private Decoding Framework

In this work, we present a more realistic and feasible design that enables an adaptable privacy level. To this end, we modify PMixED to adapt to the private and public output distributions evaluated on a given input query. Consequently, the privacy level can only be determined after the next token has been sampled. Thus we focus on two scenarios that allow us to minimize the privacy loss: Section 4.1 discusses a mechanism to filter out queries where the private and public distributions diverge extremely. (2) Section 4.2 introduces a data-dependent privacy analysis to account for queries where the private and public distributions converge. As mentioned in section 2, the high-level idea of our noisy screening and data-dependent analysis are inspired by PATE Papernot et al. (2018) but the low-level details differ substantially.

### 4.1 Noisy Screening Mechanism

The motivation is that for certain queries $\mathbf{x}$, the mixing parameter $\lambda_i$ is very small due to extreme divergence between the private and public distributions, i.e., $D_\alpha^\leftrightarrow(\overline{p}_i(\mathbf{x})||p_0(\mathbf{x}))$ is large for small $\lambda_i$. Since $\lambda_i$ is small, little information about the private data is used to assist with the prediction, leading to almost complete reliance on the public model. Perhaps, in this scenario, it would be better to save the privacy loss resulting from sampling the mixture of private and public information, and instead sample purely from the public distribution $p_0(\mathbf{x})$, which will result in no additional privacy loss since $p_0(\mathbf{x})$ was not fine-tuned on the private data $D$.

Hence, our key idea is to screen a query $\mathbf{x}$ by using the Renyi Divergence to measure the distributional difference between the projected and public output distributions. If the private and public distribution diverge extremely, we can check if mixing just a small amount of private information would already result in a large Renyi Divergence. In other words, we select a small $\lambda$, calculate $\overline{p}(\mathbf{x}) = 1/N \sum_{i=1}^{N}(\lambda p_i(\mathbf{x}) + (1-\lambda)p_0(\mathbf{x}))$, then check if $D_\alpha^\leftrightarrow(\overline{p}(\mathbf{x}_t)||p_0(\mathbf{x})) \leq T$ for some threshold value $T$. Using a small $\lambda$ also helps reduce the privacy loss from the noisy screening since less private information is used in the screening. Making this screening mechanism differentially private involves privatizing $D_\alpha^\leftrightarrow(\overline{p}(\mathbf{x}_t)||p_0(\mathbf{x}))$, which is difficult to do directly since $D_\alpha^\leftrightarrow(\overline{p}(\mathbf{x}_t)||p_0(\mathbf{x}))$ contains no upper-bound, and hence, no global sensitivity.

To work around this, we make use of the post-processing theorem of DP (Theorem A.1) by first privatizing $\overline{p}(\mathbf{x}_t)$ by adding Gaussian noise to it, $\overline{p}_{\text{priv}}(\mathbf{x}) = \overline{p}(\mathbf{x}) + \mathcal{N}(0, \sigma^2)$, then checking if $D_\alpha^\leftrightarrow(\overline{p}_{\text{priv}}(\mathbf{x}_t)||p_0(\mathbf{x})) \leq T$. However, one technical challenge is that $\overline{p}(\mathbf{x}_t)$ is a large dimensional vector, approximately 50,000 dimensions. Hence, the error of the screening mechanism scales proportionally to the size of this large vector. To reduce the noise, we use a similar approach from Tian et al. (2022); Tang et al. (2023) by selecting the top-$k$ indices $K$ of the public model $p_0(\mathbf{x})$, which does not leak additional privacy since $p_0(\mathbf{x})$ was not fine-tuned on $D$, then truncate and re-scale $p_0(\mathbf{x}_t)$ with $K$ to obtain $\overline{p}_0(\mathbf{x})$. Using $K$, we truncate $\overline{p}(\mathbf{x})$, apply noise to those $K$ indices of $\overline{p}(\mathbf{x})$, i.e., $\overline{p}_{\text{priv}}(\mathbf{x})[K] = \overline{p}(\mathbf{x})[K] + \mathcal{N}(0, \sigma^2)$, then re-scale $\overline{p}_{\text{priv}}(\mathbf{x})$. Lastly, the divergence of $\overline{p}_0(\mathbf{x}_t)$ and the DP projected distribution $\overline{p}_{\text{priv}}(\mathbf{x})$ at indices $K$ is measured.

With the noisy screening mechanism explained, we now analyze its privacy loss and defer the proof to Appendix D.

**Theorem 4.1.** Given some mixing parameter $\lambda$ and ensemble size $N$, the the noisy screening mechanism satisfies $(\alpha, \epsilon_{\text{screen}}(\alpha, N, \lambda, \sigma))$-RDP where $\epsilon_{\text{screen}}(\alpha, N, \lambda, \sigma) = \left(\frac{\lambda}{N\sigma}\right)^2 \alpha$.

### 4.2 Data-Dependent Privacy

When calculating the privacy loss of PMixED, Theorem 3.1 gives an analytical upper bound containing the Renyi Divergence order $\alpha$, the target leakage/radius $\beta$, and the ensemble size $N$. One thing to note is that this bound's derivation employs the Quasi convexity property and Triangle-like inequality of Renyi Divergence, resulting in a rather loose bound. Moreover, we observe that PMixED will answer certain queries $\mathbf{x}$ with just the private output distributions since the private and public distributions are similar. However, the data-independent privacy analysis does not account for this and instead designates a fixed privacy loss for each query, leading to overestimated privacy loss. In other words, for all $i \in [N]$, the mixing parameter $\lambda_i = 1$ yet $D_\alpha^\leftrightarrow(\overline{p}_i(\mathbf{x})||p_0(\mathbf{x})) << \beta\alpha$.

A key feature in language modeling that we can take advantage of is that LLMs inherently produce a probability distribution as its output. Thus the privacy loss of PMixED can be a direct numerical calculation of the Renyi Divergence between the output of PMixED with the entire ensemble, $p(\mathbf{x}) = \frac{1}{N} \sum_{i=1}^{N} \overline{p}_i(\mathbf{x})$, and the output of PMixED with a neighboring ensemble where the $i$-th model removed, $p_{-i}(\mathbf{x}) = \frac{1}{N-1} \sum_{j \neq i} \overline{p}_j(\mathbf{x})$. Therefore, the data-dependent privacy loss of PMixED iteratively searches through all possible neighboring ensembles to find the one with the largest privacy loss, in order to satisfy the data-dependent DP definition 3.4:

$$\epsilon_{\text{PMixED}}(\alpha, \beta, N, D, \mathbf{x}) = \max_{i \in [N]} \left\{ D_\alpha^\leftrightarrow(p(\mathbf{x})||p_{-i}(\mathbf{x})) \right\}. \quad (3)$$

Note that $\epsilon_{\text{PMixED}}(\alpha, \beta, N, D, \mathbf{x}) \leq \epsilon_{\text{PMixED}}(\alpha, \beta, N)$, since $\epsilon_{\text{PMixED}}(\alpha, \beta, N)$ is a direct analytical upper bound of $\epsilon_{\text{PMixED}}(\alpha, \beta, N, D, \mathbf{x})$. Hence, we can avoid the loose data-independent analysis by alternatively using the substantially tighter numerical, data-dependent calculation. Although

---

**Algorithm 1** `AdaPMixED`: Data-dependent PMixED with noisy screening

---

**Input.** Number of LLMs $N$, Fine-Tuned LLM's $\{p_i\}_{i=1}^N$, public model $p_0$, an input query $\mathbf{x}$, Renyi Divergence order $\alpha > 1$, target leakage $\beta$, parameters $\lambda$ and $\sigma$ for noisy screening, screening threshold $T$, top-k value $k$, current privacy loss $\epsilon$

1: $\overline{p}_{\text{priv}}(\mathbf{x}) \leftarrow \frac{1}{N} \sum_{i=1}^N \lambda p_i(\mathbf{x}) + (1 - \lambda) p_0(\mathbf{x}), \overline{p}_0(\mathbf{x}) = p_0(\mathbf{x})$
2: $K \leftarrow$ grab top $k$ indices from $p_0(\mathbf{x})$
3: $\overline{p}_{\text{priv}}(\mathbf{x})[\mathcal{V} \setminus K] \leftarrow 0, \overline{p}_0(\mathbf{x})[\mathcal{V} \setminus K] \leftarrow 0$         ▷ Truncating output distribution
4: $\overline{p}_{\text{priv}}(\mathbf{x})[K] \leftarrow \overline{p}_{\text{priv}}(\mathbf{x})[K] + \mathcal{N}(0, \sigma^2)$         ▷ Privatizing output distribution
5: Re-scale $\overline{p}_{\text{priv}}(\mathbf{x})$ and $\overline{p}_0(\mathbf{x})$ such that $\sum_{j \in K} \overline{p}_{\text{priv}}(\mathbf{x})[j] = 1$ and $\sum_{j \in K} \overline{p}_0(\mathbf{x})[j] = 1$
6: $\epsilon \leftarrow \epsilon + \left(\frac{\lambda}{N\sigma}\right)^2 \alpha$
7: **if** $D_\alpha\left(\overline{p}_{\text{priv}}(\mathbf{x})[K] \middle\| \overline{p}_0(\mathbf{x})[K]\right) > T$ **then**         ▷ Performing Noisy screening
8:     **return** $y \sim p_0(\mathbf{x}_t)$
9: **end if**
10: **for** $i \in [N]$ **do**
11:     $\lambda_i \leftarrow \arg\max_{\lambda \in [0,1]} \{D_\alpha^\leftrightarrow(\lambda p_i(\mathbf{x}) + (1 - \lambda) p_0(\mathbf{x}) \| p_0(\mathbf{x})) \leq \beta\alpha\}$
12:     $\overline{p}_i(\mathbf{x}) = \lambda_i p_i(\mathbf{x}) + (1 - \lambda_i) p_0(\mathbf{x})$
13: **end for**
14: $p(\mathbf{x}) \leftarrow \frac{1}{N} \sum_{i=1}^N \overline{p}_i(\mathbf{x})$
15: $p_{-i}(\mathbf{x}) \leftarrow \frac{1}{N-1} \sum_{j \neq i} \overline{p}_j(\mathbf{x})$
16: $\epsilon \leftarrow \epsilon + \max_{i \in [N]} \{D_\alpha^\leftrightarrow(p(\mathbf{x}) \| p_{-i}(\mathbf{x}))\}$         ▷ Performing Data-dependent analysis
17: **return** $y \sim p(\mathbf{x})$

---

the privacy loss is now a function of the private data, one can sanitize the privacy loss via smooth sensitivity analysis, similar to Papernot et al. (2018), if the privacy loss needs to be released.

### 4.3 PRIVACY AND UTILITY ANALYSIS

With our noisy screening mechanism and data-dependent accounting, we succinctly describe `AdaPMixED` in Algorithm 1 and state its privacy and utility guarantee below. The proof is left to Appendix D.

**Theorem 4.2.** Given a query $\mathbf{x}$, the next token $y$ sampled by `AdaPMixED`, Algorithm 1, is data-dependent $(\alpha, \epsilon(\alpha, \beta, N, \lambda, \sigma, D, \mathbf{x}))$-RDP where

$$\epsilon(\alpha, \beta, N, \lambda, \sigma, D, \mathbf{x}) \leq \epsilon_{\text{screen}}(\alpha, N, \lambda, \sigma) + \mathbb{1}\left\{D_\alpha^\leftrightarrow(\overline{p}_{\text{priv}}(\mathbf{x}) \| p_0(\mathbf{x})) \leq T\right\} \epsilon_{\text{PMixED}}(\alpha, \beta, N, D, \mathbf{x})$$

Our utility analysis, presented below, employs the average negative log-likelihood, which is just the logarithm of perplexity, as our utility function since our work focuses on next-token prediction. We measure the difference between the likelihood from an ensemble that only uses the private models, i.e., $\lambda_i = 1$ for all $i$, which is the best-performing ensemble, and the likelihood of the ensemble from `AdaPMixED`. To ease the analysis, we do not consider Noisy Screening.

**Theorem 4.3.** Let $\lambda_{i,t}$ denote the mixing parameter of model $i$ at query $t$ that was produced by `AdaPMixED`, $\lambda^* = J_{N,T}$ where $J$ is the $N \times T$ all-ones matrix, and $\mathcal{L}(\mathbf{x}, \lambda) = -\frac{1}{T} \sum_{t=1}^T \log \sum_{i=1}^N [\lambda_{i,t} p_i(x_t | \mathbf{x}_t) + (1 - \lambda_{i,t}) p_0(x_t | \mathbf{x}_t)]/N$ be the negative log-likelihood function. For any $T > 0$ we have the following:

$$\mathcal{L}(\mathbf{x}, \lambda) - \mathcal{L}(\mathbf{x}, \lambda^*) \leq \max_{t \in [T]} \max_{j \in [N]} \left\{(1 - \lambda_{j,t}) \log\left(\frac{p_j(x_t | \mathbf{x}_t)}{p_0(x_t | \mathbf{x}_t)}\right)\right\}.$$

Hence, we can bound the performance error from `AdaPMixED` by the likelihood of a private model $p_j(x_t | \mathbf{x}_t)$ evaluated on a query $\mathbf{x}_t$ that diverges the most from the public model $p_0(x_t | \mathbf{x}_t)$. If the divergence is large, then it causes the log-likelihood difference between the private and public model $\log(p_j(x_t | \mathbf{x}_t)/p_0(x_t | \mathbf{x}_t))$ to be large. Moreover, the likelihood of the private model with largest divergence from the public will cause the projection to be closer to the public model i.e., $\lambda_{j,t} \leftarrow 0$, increasing the utility gap. We also want to highlight that, to our knowledge, we are the first privacy-preserving decoding work that provides a utility analysis.

| Dataset | Queries Answered | Method | Privacy loss | PPL |
|---------|------------------|--------|--------------|-----|
| WikiText-103 | 1024 | Public model | $\epsilon = 0$ | $40.86 \pm 6.11$ |
| | | DP-SGD | $\epsilon = 8$ | $35.09 \pm 5.50$ |
| | | PMixED | $\epsilon = 8$ | $33.8 \pm 5.29$ |
| | | AdaPMixED | $\epsilon(D) = 0.494$ | $32.35 \pm 4.63$ |
| | 99,840 | DP-SGD | $\epsilon = 8$ | 32.53 |
| | | AdaPMixED | $\epsilon(D) = 5.248$ | 29.99 |
| One Billion Word | 1024 | Public model | $\epsilon = 0$ | $67.73 \pm 6.77$ |
| | | DP-SGD | $\epsilon = 8$ | $54.54 \pm 5.42$ |
| | | PMixED | $\epsilon = 8$ | $52.68 \pm 5.29$ |
| | | AdaPMixED | $\epsilon(D) = 0.485$ | $49.25 \pm 4.65$ |
| | 99,840 | DP-SGD | $\epsilon = 8$ | 52.97 |
| | | AdaPMixED | $\epsilon(D) = 3.186$ | 47.99 |
| PubMed | 19,968 | Public model | $\epsilon = 0$ | 32.28 |
| | | DP-SGD | $\epsilon = 8$ | 28.07 |
| | | AdaPMixED | $\epsilon(D) = 2.858$ | 26.83 |
| Air Dialogue | 16,384 | Public model | $\epsilon = 0$ | 17.33 |
| | | DP-SGD | $\epsilon = 8$ | 8.13 |
| | | AdaPMixED | $\epsilon(D) = 6.906$ | 7.67 |

Table 1: Main results of the utility-privacy tradeoff between all methods and AdaPMixED. Note that $\epsilon$ denotes data-independent privacy loss while $\epsilon(D)$ is data-dependent privacy loss. Smaller privacy loss and PPL are better.

## 5 EXPERIMENTAL EVALUTATION

### 5.1 EXPERIMENTAL SETUP

We roughly follow the same experimental setup as the original PMixED Flemings et al. (2024) on the WikiText-103 Merity et al. (2016) and One Billion Word Chelba et al. (2013) datasets. However, we experimented with two additional datasets: PubMed Cohan et al. (2018), which contains scientific papers from the medical domain, and Air Dialogue Wei et al. (2018), which contains customer service dialogue on flight booking. For models we use pre-trained GPT-2 models Radford et al. (2019) from HuggingFace Wolf et al. (2019). Since storing an ensemble of LLMs can be expensive, we opt to use LoRA Hu et al. (2021) for parameter-efficient fine-tuning. Additional details of the training and prediction hyperparameters can be found in Appendix E.

The utility is measured in terms of perplexity score (PPL), and the privacy loss $\epsilon$ is the privacy guarantee of the method after answering all queries. In the main results, 8 runs were performed and averaged to obtain the PPL and privacy loss $\epsilon$ for 1024 queries answered; for queries $> 1024$, one run was performed. 4 methods are compared in the main results: (1) a public model, which we chose to be a pre-trained GPT-2 small model; (2) DP-SGD with LoRA Yu et al. (2021b); (3) the original PMixED framework Flemings et al. (2024); (4) AdaPMixED, our method. The privacy loss is computed in RDP, but converted back to $(\epsilon, \delta)$-DP, and all methods use $\delta = 1 \times 10^{-5}$.

### 5.2 MAIN RESULTS

In this section, we highlight the privacy and utility improvement of our noisy screening mechanism and data-dependent analysis shown in Table 1. For the original datasets used in PMixED, WikiText-103 and One Billion Word, we observe that AdaPMixED outperforms the original PMixED by 1.4 PPL with 16 times less privacy. Thus, AdaPMixED not only significantly reduces the privacy loss of the original PMixED, but it also can further improve the utility. Moreover, when we increase the number of queries answered to 99,840, we observe that the privacy loss of AdaPMixED is only moderate, $\epsilon = 5.248$ and $\epsilon = 3.186$ for WikiText-103 and One Billion Word, respectively. And its utility is remarkably higher than DP-SGD, gaining 2.5 and 5.0 PPL improvement on WikiText-103

| Method | Privacy Loss: $\epsilon$ | PPL | # $\geq T$ |
|---|---|---|---|
| PMixED | 4.399 | 38.07 | 0 |
| PMixED with noisy screening | 4.139 | 38.15 | 716 |
| AdaPMixED with only Data-dependence | 0.960 | 31.42 | 0 |
| AdaPMixED | 0.924 | 31.75 | 1026 |

(a) The privacy-utility tradeoff of noisy screening and data-dependent analysis.

| Mechanism | Privacy loss: $\epsilon$ |
|---|---|
| $\epsilon_{\text{PMixED}}(\alpha, \beta, N, D, \mathbf{x})$ | 0.472 |
| $\epsilon_{\text{screen}}(\alpha, N, \lambda, \sigma)$ | 0.002 |
| RDP to DP (Theorem A.3) | 0.450 |
| Total | 0.924 |

(b) Privacy loss breakdown for AdaPMixED.

Table 2: Privacy loss and perplexity (PPL) evaluated on 9728 predictions using WikiText-103. PMixED uses $\alpha = 6$ and $\beta = 0.01$ while data-dependent PMixED and AdaPMixED uses $\alpha = 18$ and $\beta = 0.2$. Smaller values of the privacy loss $\epsilon$ is better.

and One Billion Word respectively. The results for PMixED with 100,000 queries are not shown since its utility is identical to the public model. The large query budget offers very little privacy loss for each prediction, forcing PMixED to rely heavily on the public model.

For the additional datasets, PubMed and Air Dialogue, we observe the same trend where AdaPMixED either outperforms or achieves comparable performance to DP-SGD for large-scale predictions with moderate privacy loss. These promising results demonstrate that even for substantially large inference loads, AdaPMixED obtains strong model utility while incurring a reasonable privacy loss.

## 5.3 PRIVACY-UTILITY TRADEOFF OF DATA-DEPENDENT ANALYSIS AND NOISY SCREENING

Now we investigate the privacy and utility of each component of AdaPMixED: noisy screening and data-dependent analysis. In this setup, we perform 9728 predictions on WikiText-103 using 4 methods: PMixED, PMixED with noisy screening only, AdaPMixED with data-dependent analysis only, and finally the combination of both schemes in AdaPMixED. Table 2a details the privacy-utility tradeoff of each of the 4 methods. The most evident observation here is that data-dependent analysis has the largest effect on reducing the privacy loss, a 4.5 times reduction compared to data-independent analysis. Even with PMixED employing privacy amplification by subsampling in its privacy analysis, its resulting privacy loss is still substantially larger than the data-dependent loss. Thus, the data-independent analysis pessimistically overestimates the privacy loss while data-dependent accounting provides a much tighter bound.

The effect of noisy screening on the privacy loss is smaller but still plays a significant role for both data-independent and data-dependent analysis. We observe that including noisy screening decreases the privacy loss as more queries are answered by the public model. For data-independent accounting, the additional privacy savings is 0.26 while for data-dependent it is 0.036. Additionally, we observe that noisy screening saves 0.26 privacy over PMixED while only lowering the utility by 0.08 PPL. For data-dependent analysis, noisy screening lowers the privacy loss by 0.036 with a 0.33 PPL decrease. In our view, privacy loss is a more valuable quantity than PPL, so we believe the privacy-utility tradeoff offered by our scheme is significant. Furthermore, out of the 9728 queries answered, 716 and 1026 were answered completely by the public model when using PMixED with noisy screening and AdaPMixED, respectively. Hence, this validates that a good chunk of the queries can be answered sufficiently with just the public model without diminishing the utility.

Lastly, Table 2b breaks down the privacy loss of PMixED. Noisy screening only comprises of 0.002% of the total privacy loss, clearly justifying that it saves more privacy than it loses. As expected, most of the privacy loss comes from privately decoding the next token.

## 5.4 ABLATIONS

Finally, we perform ablations on the prediction hyperparameters AdaPMixED using WikiText-103 shown in Figure 2 to better understand their privacy-utility tradeoff. We do acknowledge that AdaPMixED does contain additional hyperparameters over PMixEd and DP-SGD, although we do have fewer hyperparameters than PATE Papernot et al. (2018). However, one advantage of private

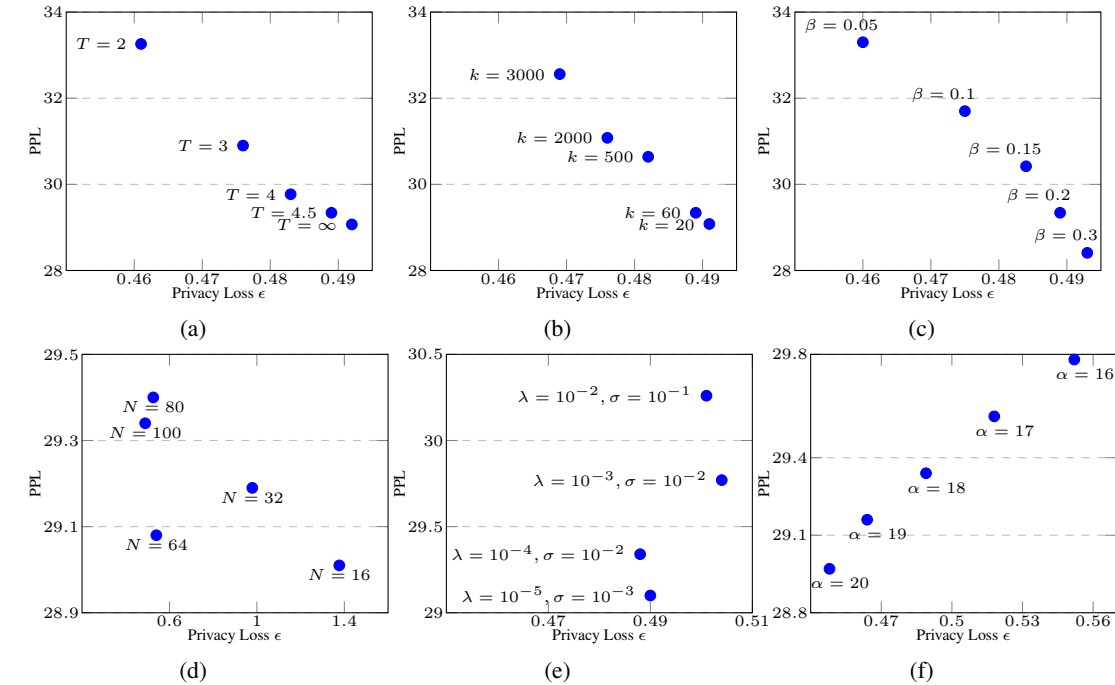

Figure 2: Ablation study on the privacy-utility tradeoff of privacy parameters, **(a)** threshold $T$, **(b)** top-$k$, **(c)** target leakage $\beta$, **(d)** Ensemble Size $N$, **(e)** noisy screening parameters $\sigma, \lambda$, and **(f)** Renyi Divergence order $\alpha$ for `AdaPMixED`.

prediction frameworks is that tuning the hyperparameters of `AdaPMixED` is considerably faster than DP-SGD because they are tuned during prediction, as opposed to retraining an LLM for the tuning process of DP-SGD.

Based on the ablation results from Figure 2, we present some rules of thumb that avoid hyperparameters tuning and still achieve reasonable privacy-utility tradeoffs. We find that $T \in [4, 4.5]$, $k \in [20, 100]$, $\beta \in [0.2, 0.3]$, $N \in [80, 100]$, $\lambda \in [10^{-4}, 10^{-5}]$, $\sigma \in [10^{-2}, 10^{-1}]$, and $\alpha \in [15, 20]$ seemingly gives a good balance between privacy and utility. Moreover, one can select values beyond these suggested bounds based on which goal– privacy or utility– one wants to prioritize. If one wants to prioritize utility, larger values of $T$ and $\beta$ should be chosen, while smaller values of $T$ and $\beta$ would help minimize the privacy loss. However, for some parameters, such as $k$, we do not suggest going beyond the bounds, as larger $k$ results in worse utility degradation and smaller privacy loss because the magnitude of additive noise from noisy screening is larger. Thus, there is an increased likelihood of exceeding the threshold $T$, resulting in purely answering the query by the public model.

# 6 CONCLUSION

In this work, we present `AdaPMixED`, a private decoding scheme that adapts to the private and public distributions evaluated on input queries to enable a large number of queries to be performed privately. Our work identified a crucial limitation of a recent work, PMixED: an analyst must set a fixed-privacy guarantee and a query budget before inference, a setup too rigid for real-world applications. The noisy screening mechanism of `AdaPMixED` filters out queries that result in large Renyi Divergence between projected and public distributions, and use the public model to answer them. The data-dependent privacy analysis performed by `AdaPMixED` exploits the output probability distribution induced by generative LLMs to directly calculate the privacy loss. These data-adaptive methods enable `AdaPMixED` to answer 100K queries while achieving reasonable data-dependent privacy loss, $\epsilon(D) = 5.248$, while still outperforming DP-SGD by nearly 2.5 perplexity. To our knowledge, no prior work has scaled private prediction methods to this large of queries while achieving strong utility. However, like all private prediction methods, eventually, the privacy loss can get too large, and hence,

one must fine-tune the LLM again on new data. But if the data received is not scarce, then it can be beneficial to do this for better privacy-utility tradeoffs. We hope that our work will spark more progress in private prediction to be a practical alternative over private training methods.

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

# A  USEFUL PROPERTIES OF RDP

**Theorem A.1** (Post-Processing Mironov (2017)). Let $A : \mathcal{D} \to \mathcal{R}$ be $(\alpha, \epsilon)$-RDP, and let $F : \mathcal{R} \to \mathcal{Z}$ be an arbitrary randomized mapping. Then $F \circ M$ is $(\alpha, \epsilon)$-RDP.

**Theorem A.2** (Composition Mironov (2017)). Let $A_1, ..., A_k$ be a sequence of $(\epsilon, \alpha)$-RDP algorithms. Then the composition $A_k \circ A_{k-1} \circ ... \circ A_1$ is $(\alpha, k\epsilon)$-RDP.

**Theorem A.3** (Conversion from RDP to Approximate DP Balle et al. (2020)). If an algorithm $A$ is $(\alpha, \epsilon)$-RDP, then it is $(\epsilon + \log((\alpha - 1)/\alpha) - (\log \delta + \log \alpha)/(\alpha - 1), \delta)$-DP for any $0 < \delta < 1$.

**Theorem A.4** (Tight Privacy Amplification by Poisson Subsampling for Renyi DP Steinke (2022)). Let $U \subseteq [n]$ be a random set that contains each element independently with probability $q$. For $x \in \mathcal{X}^n$ let $x_U \in \mathcal{X}^n$ be given by $(x_U)_i = x_i$ if $i \in U$ and $(x_U)_i = \perp$ if $i \notin U$, where $\perp \in \mathcal{X}$ is some fixed value.

Let $\epsilon : \mathbb{N}_{\geq 2} \to \mathbb{R} \cup \{\infty\}$ be a function. Let $M : \mathcal{X}^n \to \mathcal{Y}$ satisfy $(\alpha, \epsilon(\alpha))$-RDP for all $\alpha \in \mathbb{N}_{\geq 2}$ with resepect to addition or removal– i.e., $x, x^{'} \in \mathcal{X}^n$ are neighboring if, for some $i \in [n]$, we have $x_i = \perp$ or $x_i^{'} = \perp$, and $\forall j \neq i \; x_j = x_j^{'}$.

Define $M^U : \mathcal{X}^n \to Y$ by $M^U(x) = M(x_U)$. Then $M^U$ satisfies $(\alpha, \epsilon_q^{'}(\alpha))$-RDP for all $\alpha \in \mathbb{N}_{\geq 2}$ where

$$\epsilon_q^{'}(\alpha) = \frac{1}{\alpha - 1} \log \left( (1-q)^{\alpha-1}(1 + (\alpha - 1)q) + \sum_{k=2}^{\alpha} \binom{\alpha}{k}(1-q)^{\alpha-k}q^k e^{(k-1)\epsilon(k)} \right). \quad (4)$$

**Theorem A.5** (Triangle-like inequality, lemma 33.7 from Steinke (2022)). Let $P, Q, R$ be distributions on $\mathcal{R}$. If $D_\alpha(P||Q) \leq \epsilon_1 \alpha$ and $D_\alpha(Q||R) \leq \epsilon_2 \alpha$ for $1 < \alpha < \infty$, then

$$D_\alpha(P||R) \leq (\sqrt{\epsilon_1} + \sqrt{\epsilon_2})^2 \alpha. \quad (5)$$

**Theorem A.6** (Quasi-Convexity Steinke (2022)). Let $P, Q, P', Q^{'}$ be probability distributions over $\mathcal{R}$ such that $P^{'}$ absolutely continuous with respect to $Q^{'}$. For $s \in [0, 1]$, let $(1 - s)P + sP^{'}$ denote the convex combination of the distributions $P$ and $P^{'}$ with weighting $s$. For all $\alpha \in (1, \infty)$ and all $s \in [0, 1]$,

$$D_\alpha((1 - s)P + sP^{'}||(1 - s)Q + sQ^{'}) \leq \max\{D_\alpha(P||Q), D_\alpha(P^{'}||Q^{'})\}.$$

**Theorem A.7** (Convexity in Second Argument Van Erven & Harremos (2014)). For any order $\alpha \in [0, \infty]$ Renyi divergence is convex in its second argument. That is, for any probability distributions $P, Q, Q'$ and $s \in [0, 1]$

$$D_\alpha(P||(1 - s)Q + sQ') \leq (1 - s)D_\alpha(P||Q') + D_\alpha(P||Q).$$

# B  MORE BACKGROUND ON PMIXED

Algorithm 2 gives a detailed algorithmic description of PMixED.

# C  MORE DETAILS ON PRIVACY ANALYSIS OF PMIXED

The privacy analysis of PMixED is taken verbatim from the original paper Flemings et al. (2024) and is as follows: first, consider the case of no Poisson subsampling and deriving the privacy loss. Then, invoke the privacy amplification theorem A.4 to derive the final privacy guarantees. Lastly, the implications of the privacy guarantees. Let $\mathcal{P}$ denote PMixED. Note that $D$ and $D^{'}$ are neighboring datasets if $D^{'}$ adds or removes a subset $D_i$ from $D$, which is equivalent to adding or removing the model $p_i$ from the ensemble. Also recall that $\lambda_i$ is automatically chosen by solving the following optimization scheme:

$$\lambda_i \leftarrow \underset{\lambda \in [0,1]}{\arg\max}\{D_\alpha^{\leftrightarrow}(\overline{p}_i(\mathbf{x}_t)||p_0(\mathbf{x}_t)) \leq \beta\alpha\}. \quad (6)$$

---

**Algorithm 2** PMixED: A protocol for Private Next Token Prediction

---

**Input.** Number of LLMs $N$, Fine-Tuned LLM's $\{p_i\}_{i=1}^N$, public model $p_0$, total number of queries $T$, privacy budget $\epsilon_G > 0$, Renyi Divergence order $\alpha > 1$, subsample probability $0 < q \le 1$, a series of queries $\{\mathbf{x}_t\}_{t=1}^T$, Subsampled privacy loss function $\epsilon_q'(\alpha)$

1: **for** $t = 1, ..., T$ **do**
2:     Select a subset $S_t \subseteq [N]$ by choosing each model with probability $q$.
3:     $\beta \leftarrow \arg\max_{\beta'} \{\epsilon_q'(\alpha, \beta, |S_t|) \le \epsilon_G/T\}$
4:     $\lambda_i \leftarrow \arg\max_{\lambda \in [0,1]}\{D_\alpha^\leftrightarrow(\lambda p_i(\mathbf{x}_t) + (1 - \lambda)p_0(\mathbf{x}_t) \| p_0(\mathbf{x}_t)) \le \beta\alpha\} \, \forall i \in S_t$
5:     $p(\mathbf{x}_t) = p_0(\mathbf{x}_t)$
6:     **if** $S_t \ne \emptyset$ **then**
7:         $p(\mathbf{x}_t) = \sum_{i \in S_t}(\lambda_i p_i(\mathbf{x}_t) + (1 - \lambda_i)p_0(\mathbf{x}_t))/|S_t|$
8:     **end if**
9:     $y_t \sim p(\mathbf{x}_t)$
10: **end for**
11: **return** $\{y_1, ..., y_T\}$

---

**Theorem C.1.** PMixEd $\mathcal{P}$ satisfies $(\alpha, \epsilon(\alpha, \beta, N))$-RDP for some query $\mathbf{x}_t$ where

$$\epsilon(\alpha, \beta, N) \le \begin{cases} \frac{\log\left(\frac{N-1+\exp((\alpha-1)4\beta\alpha)}{N}\right)}{\alpha-1} & \text{if } N > 1 \\ \beta\alpha & \text{otherwise} \end{cases}. \tag{7}$$

*Proof.* Let $i \in [N]$ and $\mathbf{x}_t$ be a query. Define

$$p(\mathbf{x}_t) = \frac{1}{N}\sum_{i=1}^N \lambda_i p_i(\mathbf{x}_t) + (1 - \lambda_i)p_0(\mathbf{x}_t),$$

$$p_{-i}(\mathbf{x}_t) = \frac{1}{N-1}\sum_{j \ne i} \lambda_j p_j(\mathbf{x}_t) + (1 - \lambda_j)p_0(\mathbf{x}_t)$$

where $\lambda_i$ is selected from Equation 6. Now, observe that each $\lambda_i$ is dependent only on $D_i$, so $p_{-i}(\mathbf{x}_t)$ does not contain $D_i$. Using the fact that $D_\alpha^\leftrightarrow(\overline{p}_j(\mathbf{x}_t) \| p_0(\mathbf{x}_t)) \le \beta\alpha$ for all $j \in [N]$, then for any two neighboring ensembles $\{p_i\}_{i=1}^N, \{p_j\}_{j \ne i}$ with $N > 1$

$$e^{(\alpha-1)D_\alpha\left(y_t \sim \mathcal{P}(\{p_i\}_{i=1}^N, \mathbf{x}_t) \| y_t \sim \mathcal{P}(\{p_j\}_{j \ne i}, \mathbf{x}_t)\right)} = e^{(\alpha-1)D_\alpha(p(\mathbf{x}_t) \| p_{-i}(\mathbf{x}_t))}$$

$$= \mathbb{E}_{p_{-i}(\mathbf{x}_t)}\left[\left(\frac{p(\mathbf{x}_t)}{p_{-i}(\mathbf{x}_t)}\right)^\alpha\right]$$

$$= \mathbb{E}_{p_{-i}(\mathbf{x}_t)}\left[\left(\frac{\frac{N-1}{N}p_{-i}(\mathbf{x}_t) + \frac{1}{N}\overline{p}_i(\mathbf{x}_t)}{p_{-i}(\mathbf{x}_t)}\right)^\alpha\right]$$

$$\le \mathbb{E}_{p_{-i}(\mathbf{x}_t)}\left[\frac{\frac{N-1}{N}(p_{-i}(\mathbf{x}_t))^\alpha + \frac{1}{N}(\overline{p}_i(\mathbf{x}_t))^\alpha}{(p_{-i}(\mathbf{x}_t))^\alpha}\right] \tag{8}$$

$$= \mathbb{E}_{p_{-i}(\mathbf{x}_t)}\left[\frac{N-1}{N} + \frac{1}{N}\left(\frac{\overline{p}_i(\mathbf{x}_t)}{p_{-i}(\mathbf{x}_t)}\right)^\alpha\right]$$

$$= \frac{N-1}{N} + \frac{1}{N}\mathbb{E}_{p_{-i}(\mathbf{x}_t)}\left[\left(\frac{\overline{p}_i(\mathbf{x}_t)}{p_{-i}(\mathbf{x}_t)}\right)^\alpha\right]$$

$$= \frac{N-1}{N} + \frac{1}{N}e^{(\alpha-1)D_\alpha(\overline{p}_i(\mathbf{x}_t) \| p_{-i}(\mathbf{x}_t))}$$

$$\le \frac{N-1}{N} + \frac{1}{N}e^{(\alpha-1)4\beta\alpha} \tag{9}$$

where Equation 8 uses Jensen's inequality for the convex function $f(x) = x^\alpha$ since $\alpha \geq 1$ and $x \geq 0$ because we are dealing with probabilities, and Equation 9 is due to $D_\alpha(\overline{p}_i(\mathbf{x}_t)||p_0(\mathbf{x}_t)) \leq \beta\alpha$ and

$$D_\alpha(p_0(\mathbf{x}_t)||p_{-i}(\mathbf{x}_t)) \leq \max_{j \neq i} D_\alpha(p_0(\mathbf{x}_t)||\overline{p}_j(\mathbf{x}_t))$$
$$\leq \beta\alpha$$

by the Quasi Convexity property of Renyi Divergence A.6. Then using the Triangle-like Inquality A.5 gives us $D_\alpha(\overline{p}_i(\mathbf{x}_t)||p_{-i}(\mathbf{x}_t)) \leq 4\beta\alpha$. When $N = 1$, then $p_{-i}(\mathbf{x}_t) = p_0(\mathbf{x}_t)$ since PMixED will resort to using $p_0$. Hence $D_\alpha(p(\mathbf{x}_t)||p_{-i}(\mathbf{x}_t)) \leq \beta\alpha$.

Now, for the other way

$$D_\alpha(p_{-i}(\mathbf{x}_t)||p(\mathbf{x}_t)) = D_\alpha\left(p_{-i}(\mathbf{x}_t)\Big\|\frac{N-1}{N}p_{-i}(\mathbf{x}_t) + \frac{1}{N}\overline{p}(\mathbf{x}_t)\right)$$
$$\leq \frac{N-1}{N}D_\alpha(p_{-i}(\mathbf{x}_t)||p_{-i}(\mathbf{x}_t)) + \frac{1}{N}D_\alpha(p_{-i}(\mathbf{x}_t)||\overline{p}_i(\mathbf{x}_t)) \quad (10)$$
$$\leq \frac{4\beta\alpha}{N} \quad (11)$$
$$= \frac{\frac{N-1}{N}\log(1) + \frac{1}{N}\log\left(\exp\left(4\beta\alpha(\alpha-1)\right)\right)}{\alpha-1}$$
$$\leq \frac{\log\left(\frac{N-1+\exp((\alpha-1)4\beta\alpha)}{N}\right)}{\alpha-1} \quad (12)$$

where Equation 10 uses convexity in the second argument of Renyi divergence (Theorem A.7), Equation 11 uses the same argument as in Equation 9, and Equation 12 is due to concavity of logarithms. Thus $D_\alpha^{\leftrightarrow}(p(\mathbf{x}_t)||p_{-i}(\mathbf{x}_t)) \leq \epsilon(\alpha, \beta, N)$. $\qquad\square$

**Lemma C.1.** Choose $\beta$ such that

$$\beta \leq \begin{cases} \frac{\log\left(Ne^{(\alpha-1)\epsilon_G/T}+1-N\right)}{4(\alpha-1)\alpha}, & \text{if } N > 1 \\ \frac{\epsilon_G}{T\alpha}, & \text{otherwise} \end{cases}, \quad (13)$$

then $\epsilon(\alpha, \beta, N) \leq \epsilon_G/T$. Hence each prediction of PMixED satisfies $(\alpha, \epsilon_G/T)$-RDP.

**Theorem C.2.** PMixED $\mathcal{P}$ is an $(\alpha, \epsilon_G)$-RDP prediction protocol with respect to $D$.

*Proof.* Let $Q$ be an interactive query generating algorithm that generates queries $\mathbf{x}_t$. We first obtain fine-tuned weights using the training algorithm $p_i = A(p_0, D_i)$. Then $\mathcal{P}$ uses $\mathbf{x}_t$ as input and returns a response, which is a sample $y_t \sim \mathcal{P}(\{p_i\}_{i=1}^N, \mathbf{x}_t)$. Setting $\beta$ to Equation 13 guarantees that $y_t$ is $(\alpha, \epsilon_G/T)$-RDP. Then after $T$ queries and responses, the sequence $\{(\mathbf{x}_t, y_t)\}_{t=1}^T$ is $(\alpha, \epsilon_G)$-RDP by the Composition Theorem A.1. Therefore $\mathcal{P}$ is an $(\alpha, \epsilon_G)$-RDP prediction protocol. $\qquad\square$

# D    PROOF OF THEOREM 4.1, 4.2, AND 4.3

We now prove the main theorems. We restate the corresponding theorems and provide the proof details below:

**Theorem D.1.** Given some mixing parameter $\lambda$ and ensemble size $N$, the the noisy screening mechanism satisfies $(\alpha, \epsilon_{\text{screen}}(\alpha, N, \lambda, \sigma))$-RDP where $\epsilon_{\text{screen}}(\alpha, N, \lambda, \sigma) = \left(\frac{\lambda}{N\sigma}\right)^2 \alpha$.

*Proof.* We will show that the sensitivity of the noisy screening mechanism is $\Delta = \frac{\lambda\sqrt{2}}{N}$. For any two neighboring datasets $\overline{p}_{\text{priv}}(\mathbf{x}) = \frac{1}{N}\sum_{i=1}^N \lambda p_i(\mathbf{x}) + (1-\lambda)p_0(\mathbf{x})$, $\overline{p}_{\text{priv},-i}(\mathbf{x}) = \frac{1}{N-1}\sum_{j \neq i} \lambda p_j(\mathbf{x}) + (1-\lambda)p_0(\mathbf{x})$

$$\Delta = \max_{i \in [N]} ||\overline{p}_{\text{priv}}(\mathbf{x}) - \overline{p}_{\text{priv},-i}(\mathbf{x})||_2$$

$$= \max_{i \in [N]} \left\| \left( \frac{N-1}{N}\overline{p}_{\text{priv},-i}(\mathbf{x}) + \frac{1}{N}\overline{p}_i(\mathbf{x}_t) \right) - \overline{p}_{\text{priv},-i}(\mathbf{x}) \right\|_2$$

$$= \frac{1}{N} \max_{i \in [N]} \left\| \overline{p}_i(\mathbf{x}_t) - \overline{p}_{\text{priv},-i}(\mathbf{x}) \right\|_2$$

$$= \frac{1}{N} \max_{i \in [N]} \left\| (\lambda p_i(\mathbf{x}_t) + (1-\lambda)p_0(\mathbf{x}_t)) - \frac{1}{N-1} \left( \sum_{j \neq i} \lambda p_j(\mathbf{x}_t) + (1-\lambda)p_0(\mathbf{x}_t) \right) \right\|_2$$

$$= \frac{\lambda}{N} \max_{i \in [N]} \left\| p_i(\mathbf{x}_t) - \frac{1}{N-1} \sum_{j \neq i} p_j(\mathbf{x}_t) \right\|_2$$

$$= \frac{\lambda}{N}(\sqrt{2}).$$

Since the Gaussian mechanism is $(\alpha, \frac{\Delta^2 \alpha}{2\sigma^2})$-RDP, then the noisy screening mechanism is $(\alpha, \left(\frac{\lambda}{N\sigma}\right)^2 \alpha)$-RDP.

**Theorem D.2.** Given a query $\mathbf{x}$, the prediction $y$ produced by AdaPMixED, Algorithm 1, is data-dependent $(\alpha, \epsilon(\alpha, \beta, N, \lambda, \sigma, D, \mathbf{x}))$-RDP where

$$\epsilon(\alpha, \beta, N, \lambda, \sigma, D, \mathbf{x}) \leq \begin{cases} \epsilon_{\text{screen}}(\alpha, N, \lambda, \sigma) + \epsilon_{\text{PMixED}}(\alpha, \beta, N, D, \mathbf{x}), & D_\alpha^{\leftrightarrow}(\overline{p}_{\text{priv}}(\mathbf{x})||p_0(\mathbf{x})) \leq T \\ \epsilon_{\text{screen}}(\alpha, N, \lambda, \sigma), & \text{otherwise} \end{cases}$$

*Proof.* Since noisy screening is always performed, the privacy loss increases by $\epsilon_{\text{screen}}(\alpha, N, \lambda, \sigma)$. If the condition $D_\alpha^{\leftrightarrow}(\overline{p}_{\text{priv}}(\mathbf{x})||p_0(\mathbf{x})) \leq T$ is met, then the next token is sampled via PMixED, with privacy loss $\epsilon_{\text{PMixED}}(\alpha, \beta, N, D, \mathbf{x})$ from Eq. 3. So by the composition property of RDP A.2, the privacy loss will also increase by $\epsilon_{\text{PMixED}}(\alpha, \beta, N, D, \mathbf{x})$. Otherwise, the next token is sampled purely from the public distribution, hence, no further privacy is lost.

**Theorem D.3.** Let $\lambda_{i,t}$ denote the mixing parameter of model $i$ at query $t$ that was produced by AdaPMixED, $\lambda^* = J_{N,T}$ where $J$ is the $N \times T$ all-ones matrix, and $\mathcal{L}(\mathbf{x}, \lambda) = -\frac{1}{T}\sum_{t=1}^{T} \log \sum_{i=1}^{N} [\lambda_{i,t}p_i(x_t|\mathbf{x}_t) + (1-\lambda_{i,t})p_0(x_t|\mathbf{x}_t)]/N$ be the negative log-likelihood function. For any $T, N > 0$ we have the following:

$$\mathcal{L}(\mathbf{x}, \lambda) - \mathcal{L}(\mathbf{x}, \lambda^*) \leq \max_{t \in [T]} \max_{j \in [N]} \left\{ (1-\lambda_{j,t}) \log \left( \frac{p_j(x_t|\mathbf{x}_t)}{p_0(x_t|\mathbf{x}_t)} \right) \right\}.$$

*Proof.* Working through the negative log-likelihood loss, we get the following:

$$\mathcal{L}(\mathbf{x}, \lambda) - \mathcal{L}(\mathbf{x}, \lambda^*) = \left[ -\frac{1}{T}\sum_{t=1}^{T} \log \frac{1}{N}\sum_{i=1}^{N} (\lambda_{i,t}p_i(x_t|\mathbf{x}_{t-1}) + (1-\lambda_{i,t})p_0(x_t|\mathbf{x}_{t-1})) \right]$$

$$- \left[ -\frac{1}{T}\sum_{t=1}^{T} \log \frac{1}{N}\sum_{i=1}^{N} (p_i(x_t|\mathbf{x}_{t-1})) \right]$$

$$\leq \max_{1 \leq t \leq T} \max_{1 \leq j \leq N} \log p_j(x_t|\mathbf{x}_{t-1}) - \log(\lambda_{j,t}p_j(x_t|\mathbf{x}_{t-1}) + (1-\lambda_{j,t})p_0(x_t|\mathbf{x}_{t-1}))$$

$$\leq \max_{1 \leq t \leq T} \max_{1 \leq j \leq N} \log p_j(x_t|\mathbf{x}_{t-1}) - \lambda_{j,t}\log(p_j(x_t|\mathbf{x}_{t-1})) - (1-\lambda_{j,t})\log(p_0(x_t|\mathbf{x}_{t-1}))$$

$$(14)$$

$$= \max_{1 \leq t \leq T} \max_{1 \leq j \leq N} (1-\lambda_{j,t}) \log \left( \frac{p_j(x_t|\mathbf{x}_{t-1})}{p_0(x_t|\mathbf{x}_{t-1})} \right)$$

where Eq. 14 is from concavity of logarithm. Hence proves the Theorem. □

| Dataset | Method | Epochs | Learning Rate | Weight Decay | Adaptation $r$ | LoRA $\alpha$ | DP Batch Size | Clipping Norm |
|---|---|---|---|---|---|---|---|---|
| WikiText-103 | AdaPMixED | 15 | 2e-4 | 0.01 | 4 | 32 | - | - |
| | DP-SGD | 20 | 2e-4 | 0.01 | 4 | 32 | 256 | 1.0 |
| One Billion Word | AdaPMixED | 5 | 2e-4 | 0.01 | 4 | 32 | - | - |
| | DP-SGD | 9 | 2e-4 | 0.01 | 4 | 32 | 256 | 1.0 |
| PubMed | AdaPMixED | 5 | 2e-4 | 0.01 | 4 | 32 | - | - |
| | DP-SGD | 5 | 2e-4 | 0.01 | 4 | 32 | 256 | 1.0 |
| Air Dialogue | AdaPMixED | 10 | 4e-4 | 0.01 | 4 | 32 | - | - |
| | DP-SGD | 10 | 4e-4 | 0.01 | 4 | 32 | 256 | 1.0 |

Table 3: Privacy parameters used in the main results for WikiText-103[1] and One Billion World[2].

| Dataset | $\alpha$ | $\beta$ | $N$ | $\sigma$ | $\lambda$ | $T$ | Top-$k$ |
|---|---|---|---|---|---|---|---|
| WikiText-103 | 18 | 0.2 | 100 | $1 \times 10^{-2}$ | $1 \times 10^{-4}$ | 4.5 | 60 |
| One Billion Word | 18 | 0.2 | 80 | $1 \times 10^{-2}$ | $1 \times 10^{-4}$ | 4.5 | 60 |
| PubMed | 15 | 0.3 | 100 | $1 \times 10^{-1}$ | $1 \times 10^{-3}$ | 4.5 | 60 |
| Air Dialogue | 15 | 0.3 | 100 | $1 \times 10^{-2}$ | $1 \times 10^{-4}$ | 4.5 | 60 |

Table 4: Privacy parameters used in the main results for WikiText-103[1] and One Billion World[2].

# E    ADDITIONAL EXPERIMENTAL SETUP DETAILS

Training hyperparameters values used can be found in Table 3, and the prediction hyperparameters used for AdaPMixED can be found in Table 4. Each dataset was split into sequences of length 512 tokens. Our code uses existing code from the original PMixED paper Flemings et al. (2024), which is licensed by the Apache-2.0 license. We follow the terms and conditions. Our setup also follows from PMixED: certain hyperparameter values for non-private fine-tuning were selected from Hu et al. (2021), and certain hyperparameter values for private fine-tuning from Yu et al. (2021a). We employed the AdamW optimizer with weight decay 0.01 and a linear learning rate scheduler. For PMixED and AdaPMixED, the dataset is randomly partitioned into equally sized groups.

# F    LIMITATIONS

We discuss the limitations of our proposed framework. The first is that the privacy loss of AdaPMixED is data-dependent and, hence, can leak privacy if released. However, one could follow the privacy loss sanitation technique from Papernot et al. (2018) to privately release the privacy loss. The second limitation comes from the original PMixED framework, which incurs and additional inference latency due to obtaining predictions from an ensemble and calculating the mixing parameters $\lambda_i$. We also acknowledge that storing an ensemble of LLMs incurs additional storage costs, although we do employ LoRA Hu et al. (2021) to further reduce this cost. But because we do not use DP-SGD, per-example gradients do not need to be stored and ML accelerated hardware is fully utilized, thereby gaining systems performance in terms of memory and runtime. Moreover, although we showed that AdaPMixED can handle large inference loads, one advantage of DP-SGD is that it can handle an infinite number of inferences due to the post-processing theorem. Lastly, the use of a public model is not necessarily risk-free in terms of privacy Carlini et al. (2021), and thus caution is advised to further understand the privacy risks of pre-trained LLMs.

