# OpenReview forum: "Adaptively Private Next-Token Prediction of Large Language Models"
_ICLR.cc/2025/Conference — ICLR 2025 Conference Withdrawn Submission_

### Official Review · Reviewer_9tgn · 2024-11-01

**Soundness:** 2
**Presentation:** 4
**Contribution:** 3
**Rating:** 3
**Confidence:** 3

**Summary:**

The paper focuses on privacy-preserving LLMs, where the end goal is to allow untrusted parties to interact with an LLM that has been fine-tuned on a private dataset, while protecting the private dataset with differential privacy (DP). DP next-token prediction is one way of achieving this goal, which has been overlooked compared to alternatives such as DP training and fine-tuning, but is becoming increasingly relevant as most LLMs are accessed through APIs.

The paper builds on PMixED, a preexisting DP next-token prediction algorithm that mixes private next-token distributions with a public distribution, to introduce AdaPMixED. AdaPMixED uses a data-dependent privacy loss, which allows an analyst to adaptively check the privacy loss consumed so far, unlike PMixED where the privacy loss and the total number of tokens have to be fixed upfront. AdaPMixED also introduces a noisy screening mechanism, which saves privacy by routing certain queries to a public model when the private model will not help.

AdaPMixED is evaluated on four datasets (adding two datasets not evaluated by PMixED) with pretrained GPT-2 models. It achieves lower perplexity than other baselines (PMixED, DP-SGD, public model) for a significantly lower (but data-dependent!) privacy loss. This allows AdaPMixED to answer massively more queries than PMixED, which is critical to make DP next-token prediction scale and provide a credible alternative to DP training.

**Strengths:**

Main strengths:
- The privacy-utility improvements of AdaPMixED can be particularly significant, because they show that next-token prediction can be practical, compared to well-studied alternatives like DP-SGD. I was personally not aware of the PMixED line of work, and I used to consider that next-token was a dead-end in terms of DP, since privacy loss accumulates with the number of queries. It turns out that mixing public predictions with private predictions, along with a careful privacy analysis, allows such DP APIs to support pretty credible workloads (100K queries for single-digit data-dependent epsilon, whereas previous work only evaluated up to 1K queries). This is a welcome development, and I appreciate the authors' work in this important direction.
- Supporting adaptive workloads is critical, and often overlooked in the DP literature (as the authors note when referring to Dwork & Feldman). The key idea to remove the non-adaptivity limitation from PMixED is to use a data-dependent privacy loss.
- The utility results are novel and useful (Thm 4.3).

Other strengths:
- The noise-screening mechanism is original and not present in PMixED. I am hesitant to consider this technique (and Thm 4.1) as a particularly significant new result, even though the paper claims this is one of its three contributions. The adaptation seems pretty straightforward: it does not involve data-dependent loss, only some post-processing, a sensitivity calculation and a Gaussian mechanism. It can still be nice technique if it comes with experimental gains.
- The privacy loss breakdown in Tab 2b was useful.

Thank you for this paper, I learned a lot about private next-token prediction!

**Weaknesses:**

Data-dependent loss
- The data-dependent loss is a major flaw of this paper in my opinion, and the reason for my overall "reject" score. I am willing to change my score if I can be convinced that the current results are fair (maybe I misunderstood something), or with new results that incorporate a sanitized version of the data-dependent loss.
- My problem with data-dependent loss is that it can leak privacy, since it depends on the data (more specifically, it depends on the whole database passed to the DP mechanism). This does not satisfy the standard DP definition, and therefore one should not treat data-dependent epsilons and data-independent epsilons the same way. Yet, this is exactly what Table 1 is doing, by listing the "privacy loss epsilon" for DP-SGD and PMixED (meaning data-independent, with strong guarantees) and AdaPMixED (meaning data-dependent, with weak semantics) in the same table. This is simply not an apples-to-apples comparison, and any claim using this table to argue that AdaPMixED outperforms the data-independent baselines is misleading.
- The paper does acknowledge the privacy risks of data-dependent loss. But at the very least the epsilon values should then come with a clear disclaimer in the Table and summary of the results in the conclusion, or even a different notation such as $\epsilon_{data-dependent}$. This is not the case in the paper.
- The paper mentions that "one can sanitize the privacy loss via smooth sensitivity analysis, similar to Papernot et al. (2018), if the privacy loss needs to be released." This is such a crucial point. I don't think the paper can possibly afford to put this off to future work. The key results in the evaluation do release privacy loss, and without this there is no apples-to-apples comparison with data-independent baselines.


Noisy screening:
- The tradeoff evaluated in 5.3 seems pretty small. I agree that saving privacy loss is valuable, and barely changes perplexity, but the savings don't seem to be very significant. Also, I am worried that the results are falling within the uncertainty due to experimental randomness (e.g., Section 5.3 analyzes a 0.08 or 0.33 PPL decrease, but Table 1 has confidence intervals of at least 4 PPL). I think that more experiments are needed to evaluate a proper tradeoff curve with x=privacy loss and y=perplexity as the noisy screening changes, with the average of multiple runs.
- Fig 2e) attempts to do that, but it is hard to interpret since it involves 4 parameters, the values barely change, and there is no clear trend.
- Overall, it seems that noisy screening has a pretty limited impact (or at least the evaluation doesn't show clear and significant trends). This makes me doubt the importance of what is supposed to be a major contribution of the paper.

Other weaknesses and minor comments:
- The Fig 2 axis are a bit misleading, the scale should probably be fixed across all 6 experiments, if we want to compare the relative importance of different methods.
- Alg. 1 is hard to parse, having some comments or headings (e.g., "Noisy screening") would be helpful.
- PMixED has been published, you could use this reference instead of the arXiv preprint: https://aclanthology.org/2024.naacl-long.247.pdf
 - While 100K queries (where one query is one next-token prediction) is a massive improvement over previous next-token prediction work, this is only practical for a limited class of workloads. For instance, that would represent around 10 short answers (100 to 1000 tokens) per day for a few weeks on a corporate or medical dataset. Once the budget is exhausted, it's time to fine-tune the private LLMs on fresh data, which might be pretty scarce if it's only been a few weeks since the previous round of fine-tuning. This is not a limitation of the paper,  given the improvements over prior work, but it could still be interesting to discuss such practical implications.

**Questions:**

- Is there any reason why you did not analyze a sanitized version of the data-dependent privacy loss, e.g., with smooth sensitivity? I understand that there might be technical difficulties, or that smooth sensitivity could erase the gains from the data-dependent analysis. But this is a vital point in my opinion, without which it is simply not possible to compare with data-independent baselines.
- Have you considered using privacy odometers? They seem well-suited for your motivation, allowing an analyst to stop a computation adaptively and measure their privacy loss. Maybe all you need is an odometer with PMixED? If not, it could be helpful to justify why data-dependent loss is the only way forward for adaptivity.
- The noisy screening mechanism might be a good fit for the sparse vector technique, depending on how often we expect queries to skip the private ensemble.

---

> ### Author Response · Authors · 2024-11-22
> **Response by Authors Part 1**
>
> Thank you for your constructive comments. We are glad that you found the idea of the paper interesting and impactful. We address your concerns below:
>
> > My problem with data-dependent loss is that it can leak privacy, since it depends on the data
>
> This is a valid concern, however, the privacy is leaked only if the privacy parameter $\epsilon$ is disclosed. If $\epsilon$ is not disclosed then our approach will not result in a privacy leakage.
>
> > therefore one should not treat data-dependent epsilons and data-independent epsilons the same way. Yet, this is exactly what Table 1 is doing, by listing the "privacy loss epsilon" for DP-SGD and PMixED (meaning data-independent, with strong guarantees) and AdaPMixED (meaning data-dependent, with weak semantics) in the same table.
>
> Our listing of all methods under the column "privacy loss" was following the prior work on listing data-independent and data-dependent algorithms together on the main results table (Zhu et al., 2020; https://openaccess.thecvf.com/content_CVPR_2020/papers/Zhu_Private-kNN_Practical_Differential_Privacy_for_Computer_Vision_CVPR_2020_paper.pdf). Moreover, we wanted to highlight how Data-dependent analysis saves substantially more privacy over data-independent analysis, enabling large-scale private prediction that could not be done with data-independent DP. However, we do not intend to be misleading about the privacy guarantees between AdaPMixED and DP-SGD and PMixED, as we have made it clear in the methodology that AdaPMixED satisfies Data-dependent DP. Although the privacy guarantees of both approaches are not exactly comparable, we thought it would be informative to include them for utility comparision. To address your concern, we have included clear demarcation  between data-independent and data-dependent privacy loss in Table 1 and concusion.
>
> > The key results in the evaluation do release privacy loss, and without this there is no apples-to-apples comparison with data-independent baselines.
>
> Although the experimental results are "releasing" the data-dependent privacy loss in order to compare with the data-independent loss, in practice AdaPMixED does not release this value to the public. But we do acknowledge that if one wants to release the data-dependent privacy loss in a DP-manner, then it would incur additional privacy loss. Hence, practically speaking, AdaPMixED does not release the data-dependent loss, so it does not need to account for this additional privacy loss. This is not a huge practical limitation as some companies who have deployed real-world DP sytems chose not to disclose their privacy parameters. E.g., the value of $\epsilon$ in Google's safety classifier with DP synthetic data is not reported (https://research.google/blog/distributed-differential-privacy-for-federated-learning) and so does Mircosoft's Workplace Analytics which uses DP (https://download.microsoft.com/download/D/1/F/D1F0DFF5-8BA9-4BDF-8924-7816932F6825/Differential_Privacy_for_Everyone.pdf).
>
> > Overall, it seems that noisy screening has a pretty limited impact
>
> When you compare PMixED and PMixED with noisy screening on Table 2a, which shows that we can save 0.26 privacy while only degrading the utilty by 0.08 PPL, it is pretty clear that noisy screening has a substantial impact. However, when it is combined with data-dependent analysis, then the impact is not as clear mainly because data-dependent analysis has a very substantial improvement in both privacy and utility. We wanted to describe noisy screening as one key tool to deploy in the privacy-utility tradeoff approaches even if the impact in the presence of data-dependent analysis is reduced.
>
> > The Fig 2 axis are a bit misleading, the scale should probably be fixed
>
> Some parameters contribute a larger change in PPL and privacy loss compared to others. For example, changing the ensemble size from $N=16$ to $N=100$ reduces the privacy loss by nearly 1.0. No other hyperparameter has that kind of impact, and if we conform all of the axes to the same scale as the ensemble size, then we would miss out on clear trends from other parameters, even if the scale of change is much smaller. We would be happy to change the scale if that is desired by the reviewer, in spite of our above explanation.
>
> > Alg. 1 is hard to parse
>
> We have added comments to Algorithm 1 to make it easier to parse through in our revision.
>
> > you could use this reference instead
>
> We have included this reference in our revision.
>
> > but it could still be interesting to discuss such practical implications
>
> We agree and have added this discussion to the conclusion of our revision.

---

> > ### Author Response · Authors · 2024-11-22
> > **Response by Authors Part 2**
> >
> > >Is there any reason why you did not analyze a sanitized version of the data-dependent privacy loss, e.g., with smooth sensitivity?
> >
> > The challenge with smooth sensitivity is that one has to calculate the local sensitivity over all possible distances of neighboring datasets. For our data-dependent analysis, that involves calculating the data-dependent loss over all possible combinations of subsets of the enemble, which is computationally very expensive due to our limited compute availability. Hence why we left it as a future work to hopefully be solved in future iterations of AdaPMixED.
> >
> > > Have you considered using privacy odometers?
> >
> > Privacy odometers, and more relevantly renyi filters, involve individual RDP, where the privacy loss of an individual, in our case each model in the ensemble, must be a function of only the individual. Our privacy loss is a function of not just the individual, but also of the private dataset. This ability to incorporate all individuals in the private datset enables our approach to reduce privacy loss. The individual RDP (per-instance) DP definition applies more for our setup, however, it is still an open problem to apply filters (odometers) to per-instance DP. Hence we did not use privacy odometers.
> >
> > > The noisy screening mechanism might be a good fit for the sparse vector technique, depending on how often we expect queries to skip the private ensemble.
> >
> > We agree and believe it can be an interesting future work. However, the challenge, as you point out, is that we must have a good estimate on how often we expect to skip queries, which is a non-trivial task.

---

> > > ### Comment · Reviewer_9tgn · 2024-11-26
> > >
> > > Thank you for the clarifications and the updates to your submission. My minor concerns have been addressed. But two major concerns are still not addressed satisfyingly, in my opinion, regarding odometers and data-dependent privacy loss.
> > > 1. Regarding odometers, it is not true that privacy odometers (and even RDP odometer specifically) necessarily involve individual RDP. Here is an example of RDP odometer with regular RDP: https://arxiv.org/abs/2103.01379. There are other constructions, for Gaussian DP or approximate DP with advanced composition, that do not involve individual DP: https://arxiv.org/abs/2203.05481. I agree that per-instance filters/odometers seem to be an open problem, but the whole point of using an odometer would be to avoid using per-instance loss in the first place! Indeed, one of the motivations given in the paper for using data-dependent loss is to be able to stop the computation adaptively. But data-dependent privacy loss gives unacceptable privacy semantics, so I wonder if odometers can provide adaptive stopping while maintaining strong (data-independent) privacy guarantees.
> > > 2. Regarding the data-dependent loss, I appreciate the clear notation in experiments, but I am unconvinced by the argument that epsilon can be kept private in practical deployments. Differential privacy is desirable precisely because it does not operate under "security by obscurity". Early deployments, like the ones you cite, might have been hiding their privacy budgets, but this is definitely not an ideal industry practice. In fact, Google has been increasingly open about the values of epsilon (or rho for zCDP) they use (https://research.google/blog/federated-learning-with-formal-differential-privacy-guarantees/). I am not very familiar with the previous work you mention (https://openaccess.thecvf.com/content_CVPR_2020/papers/Zhu_Private-kNN_Practical_Differential_Privacy_for_Computer_Vision_CVPR_2020_paper.pdf), but they do have a caveat saying that data-dependent loss is controversial, and apparently they also include data-independent loss as well "to demonstrate that practical differential privacy can be achieved when training a deep networks under the “knowledge transfer” setting even without using data-dependent RDP.” To be clear, I am fine with data-dependent privacy loss as a tool to get a precise understanding of an algorithm, as long as the algorithm also comes with a data-independent upper bound (which Private kNN seems to be doing, and that PATE does with smooth sensitivity). This data-independent upper bound should be provided with the results and not delegated to future work.
> > >
> > > For these reasons, I am keeping my score unchanged. In fact, accepting the current submission as-is could create an additional precedent supporting the misconception that reporting results with data-dependent epsilon only is acceptable when comparing to baselines using a data-independent epsilon.

---

### Official Review · Reviewer_HAPU · 2024-11-03

**Soundness:** 3
**Presentation:** 3
**Contribution:** 3
**Rating:** 6
**Confidence:** 3

**Summary:**

The authors explore the challenges of privacy preservation in large language models (LLMs), which are typically deployed via APIs by machine learning as a service (MLaaS) providers. Traditional privacy-preserving methods, such as Differential Privacy (DP) using DP-SGD, often incur significant computational costs, and the increased noise due to privacy preservation can negatively impact the model's utility. The Private Ensemble Distribution Hybrid (PMixED) approach improves utility by mixing the output distributions of private and public models without adding noise, but it suffers from inflexible privacy constraints and does not scale well as the number of queries increases. To overcome these limitations, the authors introduce Adaptive PMixED (AdaPMixED), which dynamically optimizes the trade-off between privacy and utility based on the difference between the private and public output distributions of each query. This approach includes a novel noise filtering mechanism that filters out high-risk queries, as well as data-based privacy loss analysis, allowing for more efficient use of privacy budgets. Experiments show that AdaPMixED significantly reduces privacy loss (up to 16x compared to PMixED) while maintaining or enhancing model utility across a variety of datasets.

**Strengths:**

The author introduced several important advantages of AdaPMixED in the field of privacy protection LLM. First, AdaPMixED is highly scalable and capable of handling up to 100,000 queries with little impact on privacy and utility, which is a significant improvement over both PMixED and traditional DP methods. This makes it ideal for practical applications where MLaaS is widely used. Furthermore, this method significantly reduces privacy loss by dynamically adjusting privacy parameters based on real-time output distribution analysis. This not only saves the privacy budget but also ensures that the model’s usefulness is preserved, as evidenced by lower perplexity scores in the next token prediction task compared to other methods.

**Weaknesses:**

In my opinion, a key weakness of the AdaPMixED framework is its handling of situations where there are large divergences between private and public model outputs. In this case, the system defaults to using the output of the public model to reduce privacy risks. However, this situation illustrates that the output of private models is very important and should not be ignored. For example, the private model may provide personalized output that the public model alone cannot provide.

Furthermore, the authors primarily considered real-world usage, but the complexity of AdaPMixED may hinder its practical deployment, especially in MLaaS environments where throughput and latency are critical. The multiple decision-making layers involved in AdaPMixED, from divergent evaluation to dynamic output adjustment, can significantly increase the computational overhead. This complexity can reduce the efficiency of the system, especially under high load conditions typical in MLaaS environments. The authors could add some experiments on the trade-offs between throughput and privacy utility advantages provided by AdaPMixED.

**Questions:**

Refers to the weakness.

---

> ### Author Response · Authors · 2024-11-22
> **Response by Authors**
>
> Thank you for your constructive comments. We are glad that you found the idea of the paper interesting and impactful. We address your concerns below:
>
> > However, this situation illustrates that the output of private models is very important and should not be ignored. For example, the private model may provide personalized output that the public model alone cannot provide.
>
> We agree with the reviewer's statement and is a valid point to make. This does have strong implications for utility, however, tailoring the ensemble to provide a personalized output can lead to large privacy loss. This is one of the shortcommings of Differential Privacy (DP), not AdaPMixED, in that DP fundamentally reduces personalized outputs to prevent large reliance on private data.
>
> > The authors could add some experiments on the trade-offs between throughput and privacy utility advantages provided by AdaPMixED.
>
> The difficulty in comparing throughput between private prediction methods, such as AdaPMixED, and private training methods, such as DP-SGD, is that DP is being applied at different stages of the ML pipeline. For AdaPMixED, it is at inference time and hence the inference latency is higher when compared to DP-SGD since it does not need to apply DP at inference time. However, DP-SGD has substantially higher runtime and memory usage compared to standard training, which is what AdaPMixED uses. Thus, it is a tradeoff between training and inference latency when comparing private training vs private prediction. For extremely large language models, it might not be possible to use DP-SGD because of too high memory requirement (DP-SGD needs to store per-example gradients). However, AdaPMixED does not have this limitation and can be used with such models.

---

### Official Review · Reviewer_T9FA · 2024-11-03

**Soundness:** 2
**Presentation:** 3
**Contribution:** 2
**Rating:** 5
**Confidence:** 3

**Summary:**

This paper introduces AdaPMixED, an algorithm for differentially private (DP) next-token prediction that builds on PMixED which was introduced previously. AdaPMixED improves the practicability of PMixED, where knowing the number of queries beforehand is less likely, by dynamically adapting the privacy level based on the divergence between public and private model outputs. Specifically, AdaPMixED introduces a noisy screening mechanism to filter out queries that might incur high privacy loss, and it performs a data-dependent privacy analysis to assess and adjust privacy loss per query. Experimental results show that AdaPMixED significantly reduces privacy loss compared to PMixED while retaining strong model utility, especially in high-query scenarios.

**Strengths:**

- The paper is well-written and the evaluation includes comparing AdaPMixED with other DP methods across different datasets. The results demonstrated large privacy gains while maintaining utility, especially in high-query situations.
- The proposed method is clearly explained and simple to implement, making the previous approach PMixED more practical.

**Weaknesses:**

- The novelty is somewhat limited where it seems like only the noise screen filtering mechanism is original. The data dependent privacy analysis is an application of an existing approach to PMixED. Though the authors emphasized the importance of noise screen filtering, from the results in Table 2, majority of the privacy loss gain comes from data dependent privacy analysis.
- The model used in experiments (GPT2) seems a bit outdated, it is unclear how the privacy and utility trade-off really look like for the more recent LLMs.
- Experiment details can be explained more, e.g. how is the dataset partitioned and what is used as the public model. I understand that the setup is rather similar to the prior work PMixED but the paper should be self-contained without needing to check another paper for the setup. Some minor detailed results are missing in table 1, e.g. PMixED results are not shown for PubMed and Air Dialogue.

**Questions:**

- Is the public model the original pretrained GPT2? How do you make sure that the public model training data does not intersect with the 4 datasets in the experiment? E.g. wikipedia seems to be a rather common pretraining dataset for most public models.
- For the ablation study in Section 5, which dataset is it performed on? How does the optimal hyperparameters change with the underlying dataset?

---

> ### Author Response · Authors · 2024-11-22
> **Response by Authors**
>
> Thank you for your constructive comments. We are glad that you found the idea of the paper interesting and impactful. We address your concerns below:
>
> > The data dependent privacy analysis is an application of an existing approach to PMixED.
>
> We want to emphasize that the application of the data-dependent analysis is a key novely of our work. Because PMixED operated under the standard private prediction definition, it cannot use a data-dependent analysis. Identifying and then loosening this restriction allows us to apply a data-dependent analysis which is a critical novelty of the work. To say that our data-dependent analysis is just an application of PMixED, which did not even consider data-dependent analysis, and hence is not novel would be unfair.
>
> > The model used in experiments (GPT2) seems a bit outdated
>
> We follow the prior work's experimental setup as best as possible (see Flemings and Annavaram 2024; Yu et al., 2024; Yue et al., 2023) to give a fair comparison between AdaPMixED and DP-SGD.
>
> > Experiment details can be explained more, e.g. how is the dataset partitioned and what is used as the public model
>
> We apologize for some missing information. The public model is stated in the experimental setup (section 5.1), we used a pre-trained GPT2. However, we have included information about the partitioning in the Appendix E. In particular, we randomly partitioned the dataset into $N$ equally sized groups.
>
> > PMixED results are not shown for PubMed and Air Dialogue.
>
> The PMixED results come directly from the orignal paper (Flemings et al., 2024). Because PMixED did not experiment on PubMed and Air Dialogue, we did not include them in our results.
>
> > How do you make sure that the public model training data does not intersect with the 4 datasets in the experiment? E.g. wikipedia seems to be a rather common pretraining dataset for most public models
>
> WikiText-103 and One Billion Word are used in the original PMixED paper, so we have to experiment on these datasets in order to compare the privacy-utility improvement of AdaPMixED over PMixED. However, we do note that the original GPT2 paper (Radfor et al., 2019) did use WikiText-103 and One Billion Word in their zero-shot evaluations, so there is some reason to believe that a pre-trained GPT2 did not train on these datasets. Nevertheless, to further control for this, we added two additional datasets, PubMed and Air Dialogue, which are more modern compared to the original ones, hence even less likely to intersect with the GPT2 pre-training dataset.
>
> > For the ablation study in Section 5, which dataset is it performed on?
>
> It was performed on the WikiText-103 dataset. We have included this information in our revision.
>
> > How does the optimal hyperparameters change with the underlying dataset?
>
> We did not perform hyperparameter sweeps across all datasets. However, for our main results, we did reuse most the hyperparameters values between all datasets. And we still obtained good performance, indicating that the optimal hyperameters for one dataset can be transferrable to others.

---

### Official Review · Reviewer_5CUt · 2024-11-06

**Soundness:** 3
**Presentation:** 3
**Contribution:** 2
**Rating:** 5
**Confidence:** 3

**Summary:**

To address the limitations of differentially private (DP) training for LLMs, Adaptive PMixED (APMixED) is introduced as a private decoding framework that adapts to the divergence between private and public model outputs. It includes mechanisms to screen high-risk queries and calculate data-dependent privacy loss, achieving practical scalability and strong utility. Experimental results show that APMixED enables up to 100K predictions with reasonable privacy loss, outperforming traditional methods.

**Strengths:**

Reasonably Novel idea, that reintroduces work from PATE which was largely overlooked. Well structured and the problem is well introduced. See Questions.

**Weaknesses:**

I think that the general framing of the paper is good, but more work can be done to clarify the specific contributions and algorithms of the paper. For example, I was a bit confused by what the noisy screening method concretely is. Why is it not sparse vector, or confident-GNmax for PATE? Perhaps I missed something but it feels the algorithm could be more clearly specified.  See Questions.

**Questions:**

* I think this is more up to date with regards to tight bounds for DPSGD : https://www.usenix.org/system/files/usenixsecurity23-nasr.pdf
“We design an improved auditing scheme that yields tight privacy estimates for natural (not adversarially crafted) datasets—if the adversary can see all model updates during training”
* Can you compare for something - create private labels using this method, then train non-privately on those private labels? In order to produce a model that you can ask infinite # of queries. This is what PATE (https://arxiv.org/abs/1802.08908) does, and how it differs from CaPC  (https://arxiv.org/abs/2102.05188).

* For the empirical work, why can’t you fix epsilon? Having variable number of queries answered, privacy loss, and PPL changing makes it hard to compare. In PATE methods they tend to compare apples-to-apples, (specifically  epsilon versus label-count versus label accuracy);  like this one : https://arxiv.org/pdf/2202.10517, which isn't relevant in the specific algorithm, but rather in the presentation of epsilon versus label-count versus label accuracy.

* Did you try varying `N` partitions? In PATE this ends up being quite important.

* Do you have an assessment of how different the finetuning datasets are from the original model? The degree of dataset-similarity here matters a lot for the performance of DP finetuning algos, though in your defense you are comparing against other methods. Still it would be good to quantify this a bit better.


Overall, it seems like a good paper, but I’d like to see the algorithmic contributions to be a bit more explicitly stated in the context of existing work, and I’d like the empirical experiments to be more clearly showing their contribution.

---

> ### Author Response · Authors · 2024-11-22
> **Response by Authors**
>
> Thank you for your constructive comments. We are glad that you found the idea of the paper interesting and impactful. We address your concerns below:
>
> > For example, I was a bit confused by what the noisy screening method concretely is. Why is it not sparse vector, or confident-GNmax for PATE?
>
> Sparse vector would be an interesting future work to look into. However, the challenge is that we must have a good estimate on how often we expect the number of queries to skip the privately fine-tuned ensemble, which is a non-trivial task. And the confident-GNmax from PATE operates over histograms, voting counts of each class label. Since we are working with a different type of data, namely probability distributions, confident-GNmax does not directly apply to our setting. Hence why we needed to develop a new noisy screening mechanism that is adapted for the language modeling domain.
>
> > Can you compare for something - create private labels using this method, then train non-privately on those private labels?
>
> Yes, our method is compatible with the PATE-like setting. However, since this is the langauge modeling domain, the setup you just described where you create private labels would just be the DP synthetic data generation line of work. In this case, we would have to compare our work to other synthetic data generation methods, which is beyond the scope of our work. Our work focuses on private prediction, but can be applied to synthetic data generation. Moreover, the Pate-like setting requires access to in-distribution unlabeled public data, while AdaPMixED does not need this requirement.
>
> > For the empirical work, why can’t you fix epsilon? Having variable number of queries answered, privacy loss, and PPL changing makes it hard to compare.
>
> The main reason why we can not fix the epsilon is because the privacy loss that we are calculating is a function of private data. Hence, if we perform some sort of operation where we stop answering queries if the privacy loss exceeds some fixed epsilon value, then that operation would leak privacy. That is why we cannot fix the epsilon. However, we do fix the query budget when comparing AdaPMixED with other baselines. Hence we are comparing the privacy-utility tradeoff between AdaPMixED and other baselines under various query budgets and datasets. And our results seemingly show that AdaPMixED achieves the best privacy-utility tradeoffs.
>
> > Did you try varying N partitions?
>
> Yes, we performed ablations on the ensemble size $N$. Please look at Figure 2(d).
>
> > Do you have an assessment of how different the finetuning datasets are from the original model?
>
> One way to make this assessment is to observe the pre-trained model's performance on various datasets. "Public model" entries in Table 1 of the main results shows this comparison. Larger PPL means that the finetuning dataset has small similarity to the pre-trained model.
>
>
> > Reasonably Novel idea, that reintroduces work from PATE
>
> We would like to clarify that AdaPMixED is applied to a significantly different than PATE seting. PATE requires unlabeled public data and was intended for discriminative tasks using voting histograms in their algorithm. AdaPMixED does not use unlabeled public data and is intended for the language modeling domain, utilizing probability distributions. Hence the original techniques used by PATE do not apply to our setting.

---

### Note · Authors · 2024-12-03

**Comment:**

We have decided to withdraw the paper to improve it and address the reviewers' concerns.

**Withdrawal Confirmation:**

I have read and agree with the venue's withdrawal policy on behalf of myself and my co-authors.